# A study on the development of data technology taxonomy for data economy

**Hwasun You**⊙*, **Do-Bum Chung, Jangwon Choi, Heeseok Choi**

Korea Institute of Science and Technology Information (KISTI), Daejeon, South Korea

* hsyou@kisti.re.kr

## Abstract

As the data economy era is in full swing, the impact of data is accelerating across the economy and society. In particular, as digital transformation accelerates, data technology is becoming more important, and new products and services are being created based on data. However, despite the increasing importance of data, the lack of a comprehensive taxonomy for data technology has resulted in inadequate systematic policy development and execution. Therefore, this study proposes a data technology taxonomy that can be used for data technology-related policy making, business planning, and national R&D investment direction setting. To this end, the study defines the concept of data technology that has not been officially announced, and develops a classification system while establishing the validity of the classification system through the derivation of the limitations of similar taxonomy and the collection of expert opinions. In addition, an expert adequacy assessment will be conducted to verify whether the proposed taxonomy can be used in the field, and the current status will be analyzed by classifying the data technology-related national R&D projects based on the taxonomy. The results of this study are meaningful in helping to understand data technology and improving the system in the future when formulating data policies and conducting research and development.

## I. Introduction

Data has emerged as a new form of capital that drives industrial development and innovation growth, establishing itself as one of the most valuable assets in the world [1]. The significance of data is further emphasized by technological advancements that enable its collection, processing, analysis, and utilization. In particular, with the advent of the non-contact era triggered by the spread of infectious diseases, the digital transformation across various industry sectors and societal domains has accelerated, making data technology more crucial and fostering the creation of new data-driven products and services. In other words, the world is now moving into the data economy era, where data fuels the creation and distribution of a wide range of services throughout society.

**Data availability statement:** All relevant data are within the manuscript and its Supporting information files.

**Funding:** This research was supported by the Korea Institute of Science and Technology Information (KISTI) grant funded by the Korea government (No. K25L6M1C1). The funders had no role in study design, data collection and analysis, decision to publish, or preparation of the manuscript.

**Competing interests:** The authors have declared that no competing interests exist.

The term "data economy" was first introduced in a 2011 Gartner report by David Newman [2]. In his report, Newman (2011) argued that big data, open data, and linked data would be key to gaining a competitive advantage in the new era. The European Commission (EC) has emphasized that data is central to the future knowledge economy and society. The concept of the data economy, or data-driven economy, began to receive significant attention as a means to drive job creation and business innovation [3]. The free production, distribution, and use of data are increasingly becoming central to the economy. Many new services are emerging and expanding through the integration of data and artificial intelligence.

Data itself plays a vital role in generating economic value. It operates within an ecosystem where data is collected, shared, and utilized through interactions among producers, intermediaries, and consumers [4]. To further advance the data economy, the United States introduced the Federal Data Strategy and Action Plan in 2020 [5]. The EC released the European Data Strategy in 2020, enacted the Data Governance Act in 2022, and implemented the Data Act in 2023 [6–8].

Similarly, in 2018, South Korea transitioned to a data-driven economy, introduced several supportive policies, and passed the Data Industry Act in 2022, making it the first country to launch a comprehensive law designed to advance the data industry [9]. While countries are commonly pursuing data policies focused on legal, institutional, and governance frameworks for developing the data economy, discussions centered on data technology have not yet been fully initiated. Jack Ma (2017) emphasized the importance of the shift from Information Technology (IT) to Data Technology (DT), stating that even IT companies can become traditional businesses in the era of data technology [10]. Beom-su Kim, Chairman of Kakao (2019), also indicated that the world is rapidly transitioning from the IT business era to the DT business era, suggesting that we are currently at a turning point from the mobile era to the data technology era [11].

Market analysis firm IDC forecasts that global data volume will reach 175 zettabytes (ZB) by 2025, growing at an average annual rate of 20%. Additionally, the global market for data technology and services is projected to expand at an average annual growth rate of 18% from 2011 to 2027, with an estimated value of approximately US 103 billion dollars by 2027 [12]. Given this growth in the data economy, it is essential to build a robust foundation to support the development and use of data technology. Particularly, as data technology serves as the foundational technology that enables data to generate economic value, it should be fostered as a core technology at the national level.

Consequently, it's important to establish the foundation for creating policies, planning projects, and prioritizing national R&D investments and budget allocations for data technology. A technology taxonomy allows for assigning classification codes to development projects and technologies. This approach provides insight into the core aspects of the technologies being developed and acts as a guide for identifying related technologies during the R&D planning phase [13].

However, despite the growing emphasis on the importance of data technology, there is a lack of formalized definitions and utilization standards for data technology.

In other words, while the era of the data economy has fully commenced, the national R&D information used for establishing national R&D policies and investment strategies is managed primarily through the National Science and Technology Standard Classification System and the Future Promising New Technologies (6T) classification system. As a result, there is an insufficient framework for developing and implementing policies from the perspectives of the data economy and technology. This acts as a limitation in enhancing data technology competitiveness, necessitating discussions on the development of a data technology taxonomy. This study aims to define data technology, which has not yet been officially defined either domestically or internationally, and propose a taxonomy for it. To validate the proposed taxonomy, an expert assessment is conducted. The taxonomy is mapped to national R&D projects, and the study presents the results and discusses their implications for future research.

The purpose of this study is to define the concept of data technology and establish a data technology taxonomy. The data technology taxonomy proposed in this study is meaningful in that it can be used to set the direction of government investment related to data technology and establish new policies and strategies. In particular, the study is significant in that it focused on the perspective that the technology domain of data technology itself has grown as a result of state-led initiatives, and analyzed the status of data technology-related projects conducted as state R&D projects to check whether the proposed taxonomy can be used in the actual field. Therefore, this study aims to contribute to establishing a policy basis for supporting data technology as an independent technology system in the data economy era.

## II. Research backgrounds

### 2.1 Definitions of data technology

The concept of data technology varies among scholars. As digital transformation and advancements in technology and services in the data economy era increase the focus on data, the definition of data technology is being discussed in multiple ways. Table 1 presents the different concepts and definitions of data technology.

This table explains the concept of data technology as defined by scholars. The term data technology has not yet been clearly defined externally.

Data technology is approached from the perspective of the data lifecycle, covering the technologies necessary for collecting, processing, and managing data. Some experts define data technology in comparison to information technology. While information technology assigns meaning only to data that has been primarily processed into information, data technology considers all data to have intrinsic value. Definitions of data technology can vary and are likely to shift as technology and the environment continue to evolve. Therefore, it is necessary to synthesize these various definitions into a general concept. In this study, data technology is defined as "the technologies used in all processes that add value to data".

Table 1. Definitions of Data Technology.

| Scholar | Definition |
|---|---|
| Mohamed Ahzam Amanullah et al. [14] | Data technologies can be described as the tools or technologies that are used to efficiently process data that has been classified as data. |
| Kwak et al. [15] | Unlike IT, which only assigns value to specific pieces of important information, data technology is a universal tool that can give value to all types of data. |
| Suhyun Kim [16] | Data technology is a scientific approach that analyzes larges sets of data collected and stored through IT to discover opportunities for innovation. |
| Jack Marshall [17] Thomas H. Davenport [18] | Data technology includes solutions for data management and technologies for products or services built on data generated by people and machines. |
| Sung Hyun Park [19] | Technologies that start with measuring, collecting and storing data, and then move on to transmitting, analyzing, and interpreting it, as well as those that create information and insights from data. |

## 2.2 Limitations of previous research

To get a good grasp of the data technology landscape, it's important to look at its main aspects. Data technology mainly focuses on managing data, developing software, creating models, and making predictions about the future. As a result, its processes and outcomes are not easily visible and are often overlooked [20]. In particular, data technology requires the integration of various technologies, from IoT in the data collection stage to storage (HDFS), processing (MapReduce), and analysis (Deep Learning) [21]. Given the wide range and integration of these technologies, pinpointing the distinct and inherent features of data technology can be quite challenging.

As a result, research has shifted away from focusing solely on data technology and instead has concentrated on classifying data to understand the various components of the data ecosystem. Additionally, earlier studies often mixed up the terms 'big data' and 'big data technology' with 'data technology'. Big data itself is defined as both a technology and a category of technologies [22] and is also referred to as 'Data Intensive Technologies' [23]. The CSA report (2014) classifies big data into six dimensions necessary for building data infrastructure: data, compute infrastructure, storage infrastructure, analytics, visualization, and security & privacy [24]. Ripon Patgiri (2018) categorizes big data technologies within the context of the data lifecycle into seven distinct categories: semantic technologies, compute infrastructure, storage systems, big data management, data mining, data machine learning, and security & privacy [25]. Vitor Afonso Pinto (2020) classifies the roles within the big data ecosystem into ten categories: data creation, data acquisition, data transmission, data ingestion, data storage, data preprocessing, abstraction middleware, data analytics, data applications, and computing infrastructure [26].

There are few academic papers that specifically explore the classification of data technology. Most research typically concentrates on certain components necessary for building a data ecosystem. The data ecosystem is defined as a collection of elements that interact with each other to collect, process, store, analyze, and share data [27–29]. This data ecosystem can facilitate knowledge discovery and provide new technologies [30]. However, to effectively activate these data ecosystems and create new value through data, a supportive environment for research specifically focused on data technology is required.

Data technology acts as a facilitator that enhances organized collaboration among the key players in the data ecosystem [31]. As data travels through the value chain, its value evolves, particularly when combined with technology and when it is supported by a trust relationship with an ecosystem that includes data stakeholders [32]. Specifically, Rowley (2007) mentioned that raw data has no value until it is properly organized and stored and that data only gains value after it is transformed into knowledge [33]. In other words, data technology enhances the value of data by enabling the effective use of various types and formats.

Current research on data technology primarily focuses on the fundamental technologies required for building a data ecosystem. While the data itself is crucial, the technology used to process and analyze it is equally important. In conclusion, to understand the characteristics of data technology, optimize data usage, and drive future development through its application, it is essential to establish a clear taxonomy of data technology.

## 2.3 Necessity of data technology taxonomy system

In South Korea, the planning, evaluation, and management of national R&D projects rely on the National Science and Technology Standard Classification System and the Future Promising New Technologies (6T) classification established by the Ministry of Science and ICT. In addition, there is the ICT R&D Technology Classification, which is a classification system related to data technology.

Within the National Science and Technology Standard Classification System, the field of data technology appears to have similarities with the main category of information and communication technology (ICT) in the science and technology domain, as shown in the Table 2 below [34]. However, the classification is predominantly limited to hardware-oriented information technologies rather than comprehensively encompassing the field of data technology. This presents a significant limitation, as it lacks details on the specific technologies necessary for the application and utilization of data.

**Table 2. Data Technology Field within the National Science and Technology Standard Classification System.**

| Field | Main category | Subcategory | Sub-subcategory |
|---|---|---|---|
| Science and Technology | Artifact | ICT | Information Theory | Database, Algorithm, Information Retrieval, etc. |
| | | | Software | Embedded S/W, System Integration, etc. |
| | | | Information Security | Network System Security, Service/Application Security, etc. |

This table extracts the fields that are highly related to data technology within the National Science and Technology Standard Classification System. It shows some similarities with the general classification of information and communication in the science and technology field.

The Future Promising New Technologies (6T) classification includes IT (Information Technology), BT (Biotechnology), NT (Nanotechnology), ST (Space Technology), ET (Energy and Environmental Technology), and CT (Cultural Technology). Among these, the field most closely related to data technology is IT, as shown in Table 3. IT refers to all technologies required for the creation, extraction, processing, and storage of information throughout the entire distribution process [35]. However, the classification system is hardware-oriented and lacks categories for emerging issues such as data prediction, encryption, protection, standardization, and trading. This absence creates limitations in establishing and managing strategies for the development of related technologies.

In addition, there is the ICT Research and Development Technology Classification System approved by the Ministry of Science and ICT. This system was developed to efficiently promote the tasks related to the planning, evaluation, and management of research and development projects, and it is a classification system specialized in the information and communication and broadcasting fields [36]. Among these, the classification related to data technology is the "Big data platform" under the "Foundation SW and computing" subcategory, which is one of the 10 main categories, as shown in Table 4. The corresponding subcategories are data collection, storage, processing, management, analysis and inference, utilization, and visualization, which appear to be classified from the perspective of the data life cycle. This classification system includes the technical fields required in the data life cycle, but does not include the technical fields required for the quality control, security, and standardization of technologies, systems, or data that are the basis for utilizing, sharing, and trading data. In particular, data technology should be able to be used across all industries, but the current classification system is limited in that it is specialized for the information and communication and broadcasting sectors.

Existing similar technology classification systems fail to fully encompass the concept of data technology as defined in this study and are limited by their low formal utilization. Specifically, national R&D information such as project and performance data are investigated, analyzed, and managed solely based on the National Science and Technology Standard Classification System and the 6T classification. This presents challenges in diagnosing data technology, as well as in the development and execution of related policies. Therefore, to address recent issues such as data protection, trading, and valuation arising from technological changes, and to expand the outcomes of research projects related to data technology, systemic innovation is essential. Moreover, in a global landscape where competition in data technology is intensifying, establishing independent standards is crucial for securing technological competitiveness and leveraging data technology to uncover new opportunities.

**Table 3. Detailed Technologies in the IT Field among Future Promising New Technologies (6T).**

| Category | Technology Name |
|---|---|
| Core Components | Terabit-Class Optical Communication Component Technology, High-Density Information Storage Device, etc. |
| Next-Generation Network Infrastructure | Large-Capacity Optoelectronic System Technology, High-Speed Internet Networking Technology, etc. |
| Information Processing Systems and S/W | Multimedia Terminal and Operating System Technology, Information Retrieval and Database Technology, etc. |

This table extracts the fields that are highly related to data technology from the classification of promising new technologies (6Ts) for the future. It shows some similarities with the IT (Information Technology) field.

**Table 4. Data technology field in the ICT R&D Technology Classification System.**

| Main category | Subcategory | Sub-subcategory |
|---|---|---|
| Foundation SW and computing | Big data platform | Data collection |
| | | Storage, processing, and management |
| | | Analysis and inference |
| | | Application and visualization |

This table extracts the fields that are highly related to data technology from the ICT R&D Technology Classification System. It shows some similarities with the sub-classification of big data platforms in the underlying software and computing.

## 3. Design of data technology taxonomy

### 3.1 Design principles and methodologies

**Ethics statement.** This study did not involve human subjects research and therefore did not require ethics approval according to the research ethics guidelines of the Korea Institute of Science and Technology Information (KISTI). The study was based on expert input collected through procedures such as a technology demand survey, deliberation by a steering committee, a technical review committee, an expert validation committee, appropriateness assessments, and public opinion gathering. All data were collected within a professional context, and no personally identifiable or sensitive information was included. Accordingly, ethics approval and informed consent were waived under KISTI's institutional policies.

**Procedure.** A taxonomy is a way of organizing and grouping things based on their common features [37]. It sorts entities into groups that share similar attributes, such as their shape, topic, items, or relationships [38]. Generally, technological taxonomies are hierarchical, meaning that broader categories encompass more specific ones, with the specific categories being considered subsets or species of the broader ones. Therefore, what is true for a higher-level class is also true for its respective lower-level classes, and an entity can only belong to one class, exhibiting a mutually exclusive characteristic [39].

The data technology taxonomy proposed in this study is hierarchically classified, and methodologies for each step have been established. As shown in Table 5 below, the taxonomy was designed through several processes: reviewing existing literature and case studies, consulting an expert advisory committee, conducting a technology demand survey, validating with experts, and collecting feedback from the public. The taxonomy not only incorporates a literature review but also integrates the diverse opinions of experts in the fields of data industry, technology, and infrastructure (top-down approach). It also reflects the demand for emerging data technologies by identifying future needs through a technology demand survey (bottom-up approach).

In particular, the proposed taxonomy was designed to enhance its completeness by conducting an assessment review with experts to ensure its applicability in actual field and by gathering feedback through public briefing sessions.

In this study, six basic guidelines were established to avoid ambiguity in the taxonomy and to develop a more comprehensive system. The guidelines for designing the proposed data technology taxonomy are as follows:

1) Include both technical and non-technical activities necessary for deriving value from data.

2) Classify hierarchically, ensuring that the items within each classification level do not overlap with one another.

3) Ensure consistency in scope across technologies at the same classification level and that classification names accurately represent the corresponding technology groups.

4) Prioritize expert opinions from each field while maintaining cohesion across different fields.

**Table 5. Research Procedures and Methodology.**

| Step | Category | Descriptions |
|------|----------|--------------|
| Step 1 | Scope Definition | · Definition and scope of data technology<br>· Setting the direction for taxonomy configuration |
| Step 2 | Oversight Committee | · Review of taxonomy development principles and classification criteria<br>· Drafting of main category, subcategory and sub-subcategory |
| Step 3 | Expert Advisory Committee | · Development of the taxonomy (draft) through advisory input on various area (process, infrastructure, governance) |
| Step 4 | Technology Demand Survey | · Specific technology examination, review of redundant technologies, identification of required technologies |
| Step 5 | Technology Review Committee | · Gathering feedback on modifications, deletions, and additions to specific technologies by area |
| Step 6 | Expert Validation Committee | · Internal and external cross-validation of the taxonomy (draft)<br>· Development of revised taxonomy (draft) |
| Step 7 | Appropriateness Assessment | · Assessment of the appropriateness of the revised taxonomy (draft) |
| Step 8 | Public Feedback | · Collection of academic feedback<br>· Gathering internal and external feedback through meetings and consultations<br>· Finalization of the taxonomy |

This table provides a detailed description of the data technology classification system design process and the methodologies used at each stage. The data technology classification system was established through a process that included setting the scope of the project, a steering committee, an expert advisory committee, a technology needs survey, a technology review committee, an expert verification committee, an adequacy assessment, and a public comment period.

5) Design the taxonomy to be systematic yet simple, making it accessible to non-experts as well.

6) Aim for the taxonomy to serve as a standard for the efficient and systematic management of research and development projects, personnel, and other resources related to data technology.

## 3.2 Proposal of data technology taxonomy

This section introduces a data technology taxonomy designed to cover the technologies and activities necessary for all processes that add value to data. The taxonomy is structured considering the attributes of data technology. In particular, it is designed to include all elements required for the development of data technology, providing a basis for policy use. Through the taxonomy, research and development projects related data technology can be systematically managed. By mapping current data technologies to the taxonomy, gaps and potential future areas of data technology can be identified, providing a standard for determining the direction of research and development (R&D).

The proposed taxonomy classifies functions to meet the requirements of the data economy era, such as data technology, data infrastructure, and data utilization and dissemination. It is organized into units based on grouping according to the core technological attributes related to data (general activities, foundational technologies, management systems), and is divided into data processes, data infrastructure, and data governance.

Applying the basic guidelines for designing a data technology taxonomy summarized above, the data technology taxonomy proposed in this study is shown in Fig 1 below. The taxonomy is structured as a three-tier framework with the following levels: Main Category, Subcategory, Sub-subcategory.

When setting up the main categories, the key factor was organizing them based on the technologies, infrastructure, and legal and policy requirements needed to activate the data ecosystem. A data ecosystem refers to a system where various types of data are freely shared and utilized, acting as a catalyst for industrial development and creating innovative businesses and services [29]. Therefore, the main categories were constructed to consider not only the data itself but also all the technologies and systems that are necessary for or affected by it.

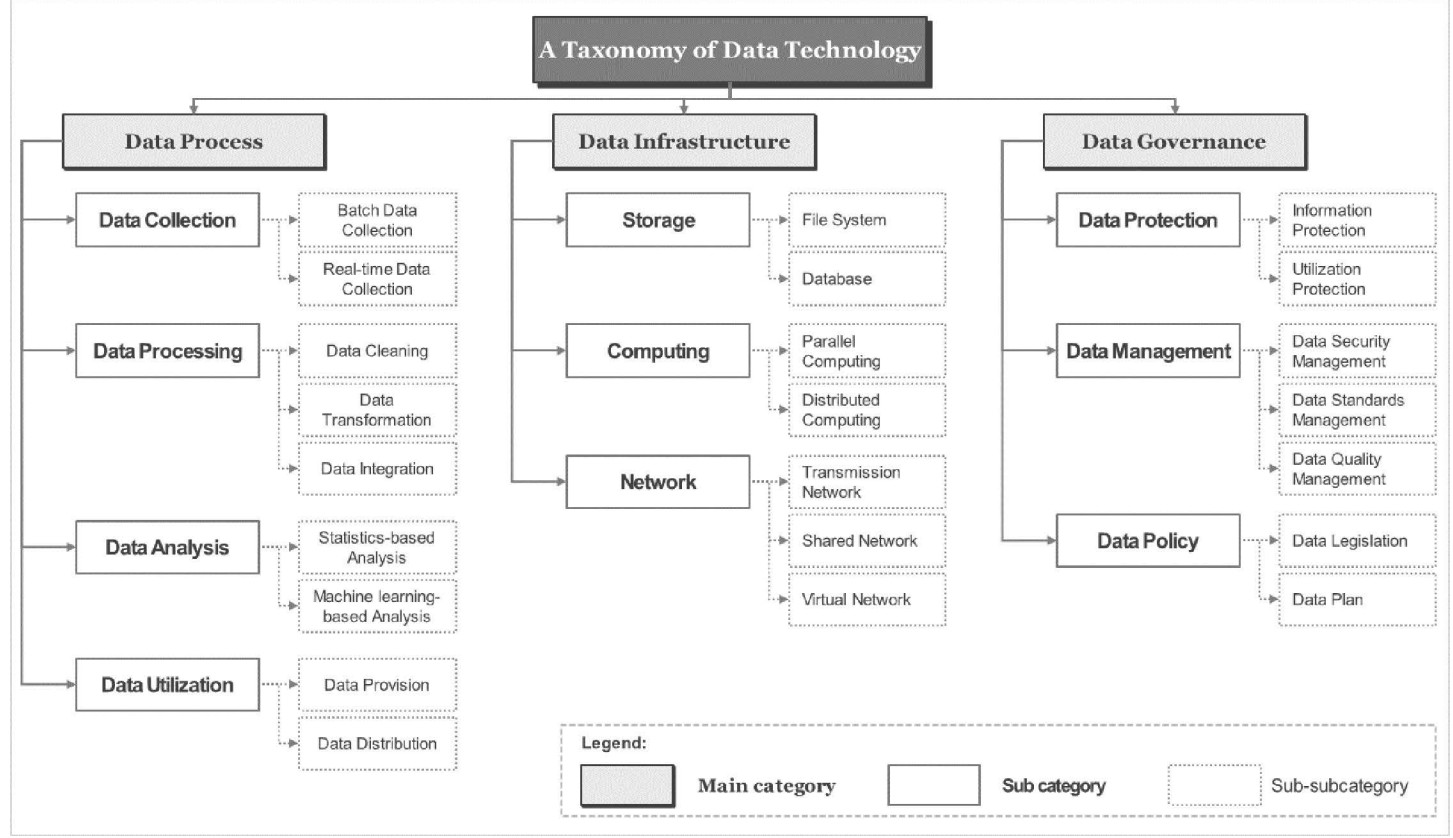

**Fig 1. A Taxonomy of Data Technology.** Fig 1. shows the results of the development of a data technology classification system. The classification system consists of a three-tier framework, with three main categories, 10 subcategories, and 23 sub-subcategories.

From the perspective of the data ecosystem, the main categories were divided into data processes, data infrastructure, and data governance. Data processes cover everything from collection to analysis and utilization. Data infrastructure involves the foundational technologies that facilitate the smooth execution of data processes, while data governance pertains to the management systems designed to enhance the value of data. Fig 2 visually presents the structure of the data technology taxonomy.

First, the main category of 'Data Process' is divided into subcategories that reflect the process of assigning value to data. These subcategories include data collection, data processing, data analysis, and data utilization. Data collection is split into batch data collection and real-time data collection based on the cycle type. Data processing is categorized into data cleansing, data transformation, and data integration, reflecting pre- and post-processing types. Data analysis is divided into statistics-based analysis and machine learning-based analysis, according to the algorithms used. Data utilization is further classified into data provision and data distribution based on the types of data dissemination activities.

Next, the main category of 'Data Infrastructure' is divided into subcategories based on the components of the data technology implementation environment. The subcategories are storage, computing, and network. Storage is further categorized into file systems and databases based on data storage methods, with a focus on whether data integrity is maintained. Computing is classified into parallel computing and distributed computing based on the method of data processing according to the characteristics of computing resources. The network is further divided into transmission network, shared network, and virtual network, depending on the purpose of interconnecting computers for data communication.

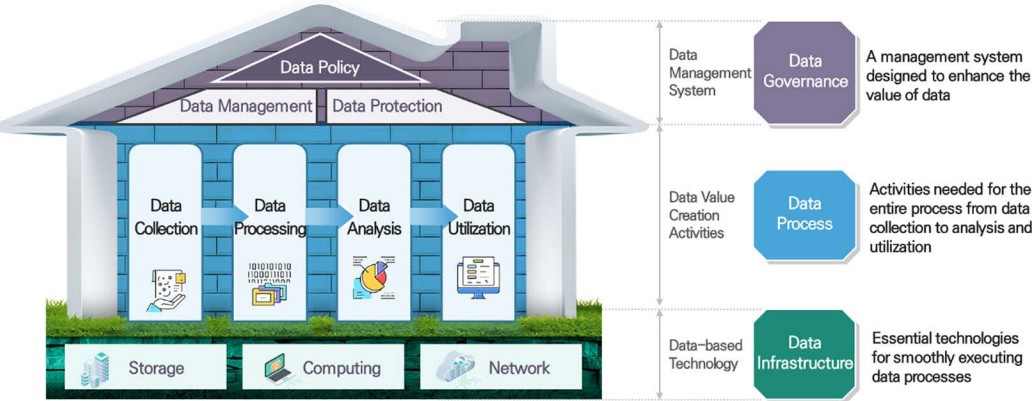

**Fig 2. Structure of the Data Technology Taxonomy.** Fig 2. shows the structure of the data technology taxonomy. To activate the data ecosystem, it has been categorized from the perspective of technology, infrastructure, and institutional development. Through this, the main category units have been divided into data processes, data infrastructure, and data governance.

Lastly, the main category of 'Data Governance' is divided into subcategories based on activities aimed at maintaining and enhancing data value. The subcategories are data protection, data management, and data policy. Data protection is categorized into information protection and usage protection based on the type of data protection technology. Data management is classified into data security management, data standards management, and data quality management, reflecting different types of data management activities. Data policy is further divided into data legislation and data planning, depending on the types of policies related to data.

Thus, the data technology taxonomy is structured with main categories, subcategories, and sub-subcategories based on their respective meanings, with their operational definitions provided below.

In this way, the subcategories and sub-subcategories of the data technology taxonomy were structured according to the main categories and their meanings. The operational definitions derived from a literature review and expert consultations are as follows. Table 6 shows the definitions of the 10 subcategories in the data technology taxonomy. Table 7 shows the definitions of the 23 sub-subcategories in the data technology taxonomy.

**Table 6. Definitions of Subcategories in the Data Technology Taxonomy.**

| Main category | Subcategory | Definition |
|---|---|---|
| Data Process | Data Collection | Activities for collecting data needed for analysis and utilization |
| | Data Processing | Activities needed to obtain data for analysis purposes |
| | Data Analysis | Activities designed to find solutions, patterns, and relationships in data |
| | Data Utilization | Activities aimed at creating added values from analyzed data |
| Data Infrastructure | Storage | Technologies needed to accumulate and store collected data |
| | Computing | Technologies needed to support quick data processing |
| | Network | Technologies that support data exchange and collaborative environments for data users |
| Data Governance | Data Protection | Activities for handling sensitive information within data and supporting its secure use |
| | Data Management | Activities needed to maintain high-quality data in a consistent and reliable fashion |
| | Data Policy | Activities involved in developing data-related legislation and planning |

This table defines each of the 10 sub categories in the data technology taxonomy. These definitions were derived from a literature review and expert consultation.

**Table 7. Definitions Sub-subcategories in the Data Technology Taxonomy.**

| Main category | Subcategory | Sub-subcategory | Definition |
|---|---|---|---|
| Data Process | Data Collection | Batch Data Collection | A way of collecting data at regular intervals |
| | | Real-time Data Collection | A way of collecting data in real-time |
| | Data Processing | Data Cleansing | A way of removing data that doesn't fit the analysis objectives from the collected data |
| | | Data Transformation | A method for transforming the shape or format of data based on specified rules while keeping its meaning intact |
| | | Data Integration | A method for integrating similar and related data to streamline data analysis |
| | Data Analysis | Statistics-based Analysis | A method for applying statistical models to refine interference and prediction |
| | | Machine Learning-based Analysis | A way of obtaining analysis results under specific conditions through data training and validation |
| | Data Utilization | Data Provision | A method designed to support data usage according to user requirements |
| | | Data Distribution | A way of evaluating data value and connecting suppliers with consumers for distribution |
| Data Infrastructure | Storage | File System | A technology for storing data in a file-centric manner based on a hierarchical directory structure |
| | | Database | A technology designed to store data in a way that allows control by a DBMS for shared use by multiple people |
| | Computing | Parallel Computing | A technology that uses multiple processing units to perform calculations quickly and simultaneously |
| | | Distributed Computing | A technology designed to handle massive computational problems by utilizing the processing power of multiple connected computers |
| | Network | Transmission Network | A technology that enables data transfer between different systems |
| | | Shared Network | A technology that allows multiple users to share existing metadata |
| | | Virtual Network | A technology that converts physical network functions into software, providing a virtual environment based on software-defined networking (SDN) |
| Data Governance | Data Protection | Information Protection | Activities aimed at protecting critical information within data such as de-identification of sensitive information and encryption |
| | | Utilization Protection | Activities focused on preventing information leaks and protecting rights during data processing and utilization |
| | Data Management | Data Security Management | Management activities targeted to ensure effective data protection |
| | | Data Standards Management | Management activities that maintain data consistency and integrity through common standards |
| | | Data Quality Management | Management activities that include diagnosis, monitoring and improvement to ensure data quality |
| | Data Policy | Data Legislation | Activities focused on establishing laws and regulations related to data protection, utilization, and management |
| | | Data Plan | Activities targeted to develop detailed implementation plans based on data-related legislation |

This table defines each of the 23 sub-subcategories in the data technology taxonomy. These definitions were derived from a literature review and expert consultation.

## 4. Applications of the taxonomy

The taxonomy proposed in the previous chapter consists of 3 main categories, 10 subcategories, and 23 sub-subcategories. Section 4.1 evaluates the suitability of the data technology taxonomy. In Section 4.2, the applicability of the taxonomy is assessed by examining its conformity with national R&D projects.

## 4.1 Assessment of taxonomy's suitability

To assess the practical applicability of the proposed data technology taxonomy and enhance its validity for use, an evaluation was conducted with 23 experts in the field of data technology. The evaluation criteria and items were selected with reference to the guidelines of the National Science and Technology Standard Classification System in South Korea [40]. The evaluation criteria consist of five categories: comprehensiveness, independence, universality, policy relevance, and usability. The meanings of each criterion are outlined in Table 8 below. The scoring criteria for each evaluation criterion are detailed in Table 9 below.

The expert evaluation was conducted based on a 5-point scale for each item. The results showed that comprehensiveness was evaluated with a score of 4.3, independence with a score of 4.6, universality with a score of 4.6, policy relevance with a score of 4.4, and usability with a score of 4.3. Therefore, the average score of 4.4 across all five categories, with each individual category receiving a score of 4 or higher. This indicates that the taxonomy is generally well-constructed. In other words, including the results of the evaluation, it was confirmed that the proposed data technology taxonomy aligns with the reasonably established design principles outlined earlier.

## 4.2. Matching with national R&D projects and results

This study examined national R&D projects to see how effectively the data technology taxonomy from the previous chapter functions in practice. One goal of creating this taxonomy is to provide a classification standard that can identify the status of government R&D investments by data technology type and manage related projects more efficiently. Thus, the study aimed to categorize national R&D projects in data technology using the developed taxonomy.

To match national R&D projects with the data technology taxonomy, the National Science & Technology Information Service (NTIS) was utilized. NTIS is a national R&D information knowledge portal in South Korea that provides information

**Table 8. Evaluation Criteria for Data Technology Taxonomy Suitability.**

| Category | Descriptions and Considerations |
|---|---|
| Comprehensiveness | ○ **The extent to which the proposed taxonomy covers all aspects related to data technology, in accordance with the definitions of the data technology taxonomy framework**<br>* Definitions of data technology taxonomy framework: Includes all technologies and activities necessary for the processes that add value to data |
| Independence | ○ **The extent to which items within the same category are independent and distinct from one another**<br>* Considers whether there is any overlap or redundancy among items when comparing them within the main category, subcategory and sub-subcategory |
| Universality | ○ **The extent to which the proposed taxonomy conforms with the general characteristics of other classification systems** (e.g., the National Science & Technology Standard Classification System, etc.) **in terms of hierarchical appropriateness and classification criteria suitability**<br>* Universality: Represents the similarity and consistency in the structure and characteristics that are fundamentally present in classification systems across various fields |
| Policy Relevance | ○ **The extent of conformity with national science and technology policies (plans) and data-related policies**<br>* Considers how much the taxonomy contributes to implementing data-related policies, the urgency of its adoption and the impact of its implementation |
| Usability | ○ **The extent to which government officials, researchers, and other users can apply and utilize the taxonomy easily from their perspective**<br>* Considers how the taxonomy can be used for developing data-related policies, planning new projects, and identifying R&D opportunities |

This table shows the items for reviewing the adequacy of the data technology taxonomy. The items for evaluating the adequacy were selected by referring to the guidelines for the National Science and Technology Standard Classification System in Korea, and the evaluation items are five: comprehensiveness, independence, universality, policy relevance, and usability. The meaning of each item and the considerations for evaluation are summarized below.

**Table 9. Scoring Criteria for Data Technology Taxonomy Evaluation Items.**

| Score Criteria Table | | | | | | |
|---|---|---|---|---|---|---|
| Category | | Score | | | | |
| | | 1 | 2 | 3 | 4 | 5 |
| 1 | Comprehensiveness | Low | <------------------------> | | | High |
| 2 | Independence | Low | <------------------------> | | | High |
| 3 | Universality | Low | <------------------------> | | | High |
| 4 | Policy Relevance | Low | <------------------------> | | | High |
| 5 | Usability | Low | <------------------------> | | | High |

This table shows the scoring criteria for each item in the data technology taxonomy. The evaluation is based on a five-point scale, and the higher the number, the higher the score for the evaluation item.

on projects, tasks, personnel, and research and development outcomes related to national R&D projects. National R&D projects refer to those funded by budgets or grants for research and development, provided according to laws and regulations set by central administrative agencies. The government views R&D as crucial for driving technological innovation and ensuring future growth. In 2024, the government's R&D budget even exceeded KRW 26.5 trillion won.

The national R&D projects in data technology focused on tasks from 2018 to 2022. To build the dataset, a keyword-based search query was developed for the NTIS database, and project titles were reviewed by experts in related fields. The search process initially extracted projects using the keyword "data", followed by a thesaurus operation based on the initial results. Finally, projects were matched by constructing keywords based on the sub-subcategories of the data technology taxonomy.

The Fig 3 below visualizes national R&D projects classified according to the data technology taxonomy. It shows the distribution of national R&D projects related to data technology and enables the identification of characteristics and evolutionary directions in data technology development by matching national R&D projects with the data technology taxonomy. All relevant national R&D projects were fully matched to the data technology taxonomy, and the matching results are presented based on national R&D investment amounts.

Among the national R&D projects related to data, those classified under "Data Process" in the main categories of the taxonomy accounted for approximately 70.2% of the total investment, highlighting the government's strong emphasis on funding this area. Next, "Data Governance" represented about 20.2% of the investment, while "Data Infrastructure" accounted for 9.6%. This distribution indicates that national R&D projects related to data are more focused on technologies for processing and utilizing data using infrastructure rather than on projects for building or operating the infrastructure.

The above results were further analyzed by main category and by year to examine the classification of national R&D projects. In the field of data process, the number of projects has consistently increased over the past five years, with an average annual growth rate of 19%. In particular, projects related to data analysis and data utilization made up more than 66.5% of the data process category, indicating that national R&D projects are heavily concentrated in these research areas. Although the data infrastructure category has relatively fewer matched projects in terms of investment, it has been growing at an average annual growth rate of 16%, with a notable increase in projects related to distributed computing. The data governance category has also seen an average annual growth rate of 19%, with recent investments increasingly targeting data protection and data management projects.

These results suggest that national R&D investment is primarily focused on technologies within the data process category. The trends in domestic research and development related to data technology indicate a strong emphasis on data analysis, ensuring that collected or processed data is used effectively, and on data utilization, where raw or analyzed data

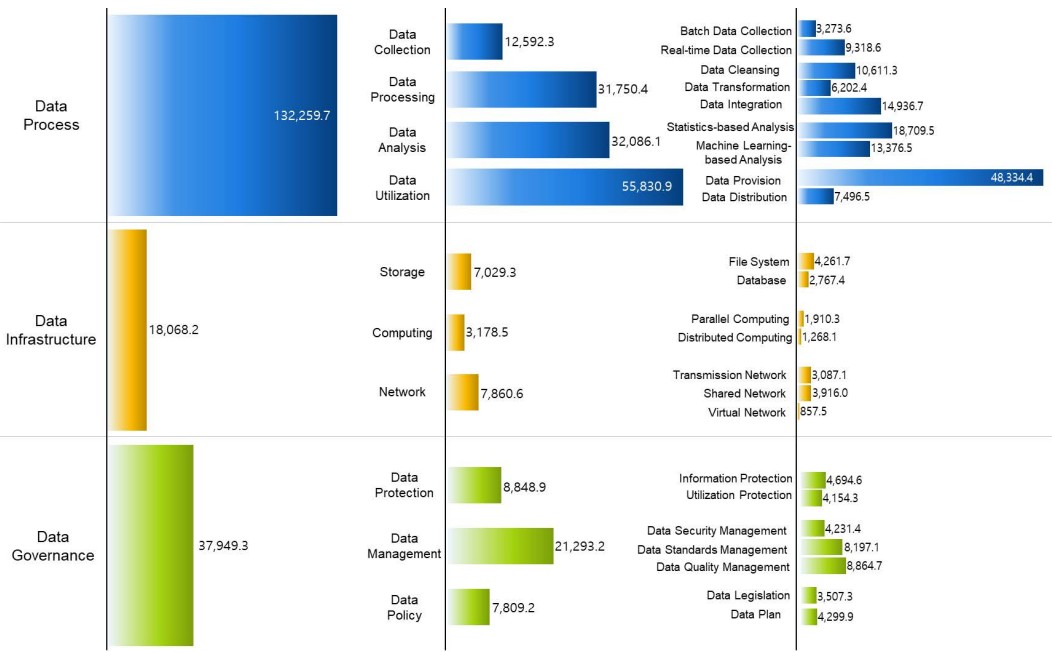

**Fig 3. National R&D Funding by Category (Unit: x KRW 100 million).** Fig 3. is a visualization of national R&D projects classified according to the data technology taxonomy. It is presented based on the amount of national R&D investment, and the figure shows the distribution of national R&D projects related to data technology.

is applied to products or services. Additionally, there has been a recent rise in projects related to data management and data protection, aimed at improving data quality and usability. These findings are likely connected to the government's announcement of the "Data Protection Core Technology Development Strategy". This suggests that the country is adopting a strategy of concentration, with substantial investment in specific research areas.

## 5. Conclusion

The data technology taxonomy proposed in this study, along with the case studies, leads to the following conclusions: The taxonomy is designed to include all technologies involved in enhancing the value of data throughout various processes, as previously defined. It also takes into accounts its applicability to policy and practical use.

This study developed a data technology taxonomy, validated it qualitatively through expert evaluations, and confirmed its practical applicability by aligning it with national R&D projects through quantitative analysis. The verification results are as follows: The taxonomy received a generally favorable evaluation based on the aggregated scores from the appropriateness assessment. Additionally, its application to data-related R&D projects was thoroughly validated, demonstrating its strong potential for practical use.

Therefore, the data technology taxonomy proposed in this study is meaningful in that it can guide national R&D investments and inform policy decisions. As data technology continues to advance, there is a growing limitation in that technologies not encompassed within the National Science and Technology Standard Classification System and the 6T classification lack a basis for conducting R&D. Therefore, a policy basis to support data technology as an independent technological system must be established. The data technology taxonomy can serve as a standard for identifying and securing core data technologies and developing strategies for acquiring new ones by creating a data technology roadmap based on the taxonomy. Moreover, the data technology taxonomy could serve as a critical tool for the data industry

economy, impacting areas like data trading, valuation, platform support, and data scientist training. To fully realize its potential, it should be formalized as an essential element of the data-driven economy.

## 6. Limitations of the study and future research directions

This study has developed a data technology taxonomy and conducted case studies. The final proposed data technology taxonomy not only reflects fragmented technologies, but also aims to increase its usability by including all technologies used in the process of adding value to data, as defined in the previously defined data technology concept. Therefore, the data technology taxonomy covers both technical and non-technical activities necessary to create value from data throughout the data life cycle. From this perspective, the data technology taxonomy is unique. However, this study has the following limitations.

First, the data technology taxonomy has not yet been fully implemented, so there may be other assessments of completeness. Of course, in order to design a highly complete taxonomy, expert adequacy reviews, public opinion surveys, and application to national R&D projects were conducted. However, when applied in the actual policy process, a certain degree of subjective interpretation may be involved, which may raise concerns about the completeness of the proposed taxonomy.

Second, although data technology has been defined, it is not a concept that has been publicly announced. Therefore, the approach may vary depending on the subjectivity inherent in its interpretation. Researchers may perceive data technology differently, which may lead to various interpretations and applications.

Finally, there is the issue of the open use of the taxonomy in terms of utilization. The establishment of a data technology taxonomy facilitates the understanding of the current state of technology development and the government's R&D investment. It is judged that the government's official approval will be required to expand the use of the data technology taxonomy. Therefore, continuous research is needed to improve the system so that it can be recognized as an official technology taxonomy by the government. It will be necessary to reflect the opinions on improving the data technology taxonomy through public presentations of research results and to promote it.

This study aims to lay the foundation for future research directions by acknowledging these limitations and considerations, and to lay the foundation for future research directions on the applicability of the data technology taxonomy. Future research should specify the detailed technologies within the data technology taxonomy, apply the technology taxonomy developed for specific tasks related to actual data technologies, and conduct quantitative analysis of the effects of R&D investment. This will enable us to predict promising new technologies of the future and identify technologies that require intensive support, and we believe that strategic investment in each data technology sector will enable us to secure data technology in a preemptive manner. In addition, gradual application research is needed to ensure that the data technology taxonomy can be used and spread as an essential element of the data economy.

## Supporting information

**S1 Data. The raw numerical data used to generate Fig 3. It includes the amount of national R&D investment classified by the proposed data technology taxonomy.**
(XLSX)

## Author contributions

**Data curation:** Hwasun You.

**Formal analysis:** Hwasun You.

**Investigation:** Hwasun You.

**Methodology:** Hwasun You.

**Supervision:** Do-Bum Chung, Jangwon Choi, Heeseok Choi.

**Writing – original draft:** Hwasun You.

**Writing – review & editing:** Hwasun You, Do-Bum Chung, Jangwon Choi, Heeseok Choi.

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
