## [Decision Letter · Decision Letter 0]

PONE-D-25-03564A Study on the Development of Data Technology Taxonomy for Data EconomyPLOS ONE

Dear Dr. You,

Thank you for submitting your manuscript to PLOS ONE. After careful consideration, we feel that it has merit but does not fully meet PLOS ONE’s publication criteria as it currently stands. Therefore, we invite you to submit a revised version of the manuscript that addresses the points raised during the review process.

**Kindly focus on issues related to the paper structure and flow of your work. All comments are attached with this email.**

**For figures request, you might place the figures inside one manuscript to see how it will look (pending acceptance).**

A rebuttal letter that responds to each point raised by the academic editor and reviewer(s). You should upload this letter as a separate file labeled 'Response to Reviewers'.A marked-up copy of your manuscript that highlights changes made to the original version. You should upload this as a separate file labeled 'Revised Manuscript with Track Changes'. (highlighted in yellow, with figures inserted)An unmarked version of your revised paper without tracked changes. You should upload this as a separate file labeled 'Manuscript'.

We look forward to receiving your revised manuscript.

Kind regards,

Issa Atoum

Academic Editor

PLOS ONE

Journal Requirements:

This research was supported by the Korea Institute of Science and Technology Information (KISTI) grant funded by the Korea government (No. K25L6M1C1-01).

6. Please ensure that you refer to Figure 1, 2, and 3 in your text as, if accepted, production will need this reference to link the reader to the figure.

7. We note you have included a table to which you do not refer in the text of your manuscript. Please ensure that you refer to Table 2, 3, 4, 5, 6, 7, and 8 in your text; if accepted, production will need this reference to link the reader to the Table.

Additional Editor Comments :

**Kindly spare one section for discussion and limitation or threats to validity, and future research (already available in different sections of the paper).**

Reviewers' comments:

Reviewer's Responses to Questions

**Comments to the Author**

1. Is the manuscript technically sound, and do the data support the conclusions?

Reviewer #1: Yes

2. Has the statistical analysis been performed appropriately and rigorously? 

Reviewer #1: Yes

3. Have the authors made all data underlying the findings in their manuscript fully available?

Reviewer #1: Yes

4. Is the manuscript presented in an intelligible fashion and written in standard English?

Reviewer #1: No

5. Review Comments to the Author

Reviewer #1: 

The paper presents novel taxonomy of the data technology. The article and contributions seems to be novel but are not presented well. So I suggest a Major revision for this article.

Here are my comments:

The figures are not presented in the paper. It is better to add the diagrams in the paper rather than adding them separately in the submission.The novel contributions of the paper should be presented at the end of introduction section but the paper does not have contributions mentioned anywhere.The uniqueness of the work should be presented in the related work section recommended to be at the end of the related work section.The limitations of the paper are not presented. Limitations section help the researchers in the community to find the gaps and come up with ideas that can fulfill the gap.The formatting of the paper is not appropriate. Please check the guidelines and use correct formatting.The comparison with the other taxonomies in the same field is absent. It should present a very good comparison between the papers published in the same domain.I recommend the authors to use Latex for writing as the formatting will become simple and easier. It is very tough to follow the paper at the present stage.It is recommended for the authors to present the guidelines given for the experts in separate section with highlighted subheading.

6. PLOS authors have the option to publish the peer review history of their article (what does this mean? ). If published, this will include your full peer review and any attached files.

**Do you want your identity to be public for this peer review?** For information about this choice, including consent withdrawal, please see our Privacy Policy .

Reviewer #1: **Yes: **

---

## [Author Response · Author response to Decision Letter 1]

18 May 2025

< Journal Requirements:>

We feel that it has merit but does not fully meet PLOS ONE’s publication criteria as it currently stands. Therefore, we invite you to submit a revised version of the manuscript that addresses the points raised during the review process.

(Response)

Thank you for your positive feedback. We have reviewed the entire manuscript and corrected errors in file names and manuscript format. Additionally, we have described the role of the funding provider in the cover letter.

Furthermore, this study does not constitute human subjects research and therefore does not require ethics approval or participant consent under the research ethics guidelines of the Korea Institute of Science and Technology Information (KISTI). Accordingly, to clarify the ethical standards of the study, we have added an Ethics Statement at the beginning of Section 3.1 (Design Principles and Methodologies) in the manuscript. This revision was made to ensure full compliance with PLOS ONE’s ethical reporting requirements.

Figures 1, 2, and 3 have been mentioned and referenced in the main text. Additionally, Tables 1, 2, 3, 4, 5, 6, 7, 8, and 9 have been referenced in the main text. Captions have been added to the references at the end of the manuscript, and citations within the text have been revised accordingly.

We have revised the manuscript overall to reflect the journal requirements. We sincerely appreciate the valuable feedback provided to improve this manuscript.

< Additional Editor Comments >

1. Kindly spare one section for discussion and limitation or threats to validity, and future research (already available in different sections of the paper).

(Response)

Thank you for pointing out the limitations of this manuscript. In accordance with the editor's opinion, we have added a separate section to the paper. We have added a section titled “Limitations of the study and future directions,” which provides a detailed discussion of the validity of the study, its limitations, and future research directions.

< Reviewers' comments>

1. The figures are not presented in the paper. It is better to add the diagrams in the paper rather than adding them separately in the submission.

(Response)

Figures 1, 2, and 3 have been added to the paper, and references have been created.

2. The novel contributions of the paper should be presented at the end of introduction section but the paper does not have contributions mentioned anywhere.

(Response)

A new contribution to the paper has been added at the end of the introduction section.

The data technology taxonomy proposed in this study is meaningful in that it can be used to set the direction of government investment related to data technology and establish new policies and strategies. In particular, the study is significant in that it focused on the perspective that the technology domain of data technology itself has grown as a result of state-led initiatives, and analyzed the status of data technology-related projects conducted as state R&D projects to check whether the proposed taxonomy can be used in the actual field. Therefore, this study aims to contribute to establishing a policy basis for supporting data technology as an independent technology system in the data economy era.

3. The uniqueness of the work should be presented in the related work section recommended to be at the end of the related work section.

(Response)

Regarding the uniqueness of the paper, we have added a section titled “6. Limitations of the study and future research directions.”

The final proposed data technology taxonomy not only reflects fragmented technologies, but also aims to increase its usability by including all technologies used in the process of adding value to data, as defined in the previously defined data technology concept. Therefore, the data technology taxonomy covers both technical and non-technical activities necessary to create value from data throughout the data life cycle. From this perspective, the data technology taxonomy is unique.

4. The limitations of the paper are not presented. Limitations section help the researchers in the community to find the gaps and come up with ideas that can fulfill the gap.

(Response)

The limitations of this paper have been added in the section titled ”6. Limitations of the Study and Future Research Directions.”

To mention briefly, First, the data technology taxonomy has not yet been fully implemented, so there may be other assessments of completeness. Second, although data technology has been defined, it is not a concept that has been publicly announced. Finally, there is the issue of the open use of the taxonomy in terms of utilization. The establishment of a data technology taxonomy facilitates the understanding of the current state of technology development and the government's R&D investment.

5. The formatting of the paper is not appropriate. Please check the guidelines and use correct formatting.

(Response)

The paper format has been revised according to the guidelines. Specifically, reference materials have been added for each Table 1, 2, 3, 4, 5, 6, 7, 8, 9, and Figure 1, 2, 3. Additionally, captions have been added to the references, and all in-text citations have been revised accordingly.

6. The comparison with the other taxonomies in the same field is absent. It should present a very good comparison between the papers published in the same domain.

(Response)

A comparison with other classification systems in the same field has been added to Section 2.3 Necessity of Data Technology Taxonomy System for further analysis. Although there are few classification systems in the same field, comparing with the ICT R&D Technology Classification System has further strengthened the validity of establishing a data technology classification system. Thank you for your feedback.

7. I recommend the authors to use Latex for writing as the formatting will become simple and easier. It is very tough to follow the paper at the present stage.

(Response)

In accordance with the reviewer's comments, we will use Latex to write future papers. Thank you for the useful information.

8. It is recommended for the authors to present the guidelines given for the experts in separate section with highlighted subheading.

(Response)

During the revision process, we added a section titled “6. Limitations of the Study and Future Directions” to incorporate the reviewer's comments.

The future research should specify the detailed technologies within the data technology taxonomy, apply the technology taxonomy developed for specific tasks related to actual data technologies, and conduct quantitative analysis of the effects of R&D investment. This will enable us to predict promising new technologies of the future and identify technologies that require intensive support, and we believe that strategic investment in each data technology sector will enable us to secure data technology in a preemptive manner. In addition, gradual application research is needed to ensure that the data technology taxonomy can be used and spread as an essential element of the data economy.

We have revised the manuscript overall to reflect the reviewer's comments. We sincerely appreciate your valuable feedback, which has been instrumental in improving the manuscript and enhancing its quality.

---

## [Editor Report · Decision Letter 1]

A Study on the Development of Data Technology Taxonomy for Data Economy

PONE-D-25-03564R1

Dear Dr. You,

We’re pleased to inform you that your manuscript has been judged scientifically suitable for publication and will be formally accepted for publication once it meets all outstanding technical requirements.

Kind regards,

Issa Atoum

Academic Editor

PLOS ONE
---

## [Editor Report · Acceptance letter]

PONE-D-25-03564R1

PLOS ONE

Dear Dr. You,

I'm pleased to inform you that your manuscript has been deemed suitable for publication in PLOS ONE. Congratulations! Your manuscript is now being handed over to our production team.

Kind regards,

on behalf of

Dr. Issa Atoum

Academic Editor

PLOS ONE